# Body composition, physical fitness and physical activity in Mozambican children and adolescents living with HIV

Nivaldo Chirindza[1]*, Lloyd Leach[2], Lucília Mangona[3], Gomes Nhaca[1], Timóteo Daca[1], António Prista[1]*

1 Physical Activity and Health Research Group, FEFD, Universidade Pedagógica de Maputo, Maputo, Mozambique, 2 Department of Sport, Recreation and Exercise Science, University of the Western Cape, Cape Town, South Africa, 3 School of Sport Sciences, Universidade Eduardo Mondlane, Maputo, Mozambique

* aprista1@gmail.com (AP); nivaldochirindza@yahoo.com.br (NC)

**Data Availability Statement:** All relevant data are within the article and its Supporting Information files.

## Abstract

### Introduction

As a result of the effectiveness of antiretroviral drugs (ART), HIV/AIDS has become a chronic disease, which has enabled children living with HIV to reach adolescence and adulthood. However, the long exposure to both the disease and ART has caused undesirable effects that compromise the physiological functioning and the quality of life of the subjects.

### Objective

To determine the body composition, physical fitness and habitual physical activity of children and adolescents living with HIV on ART.

### Methods

A total of 79 subjects of both genders aged 8–14 years, living with HIV in ART, selected by convenience participated in the study. The subjects underwent anthropometric assessment, physical fitness assessment and physical activity assessment.

### Results

Relative to reference norms, the values of the anthropometric indicators fell below 50th percentile (height/age = 92.4%; BMI/age 72.2%; sum of skinfolds = 51.9%; arm circumference = 63.3%). The prevalence of "low height/age" and "low weight/age" was 34.9% and 9.3%, respectively for boys, and 27.9% and 11.1%, respectively for girls. With the exception of trunk flexibility (12.3%), most subjects were considered unfit in the physical fitness tests (abdominal resistance = 76.4%; handgrip strength = 75.4%; lower limb power = 66.4%). The percentage of subjects with insufficient physical activity was 45.5% for boys and 77.8% for girls. The values for all variables were consistently and significantly lower when compared with studies done in Mozambicans boys and girls without HIV+ from both urban and rural areas.

**Funding:** We have not any grant, sponsorship or support from our institution to pay the publication we are requesting fee waive for the publication

**Competing interests:** All data generated or analyzed during this study are included in this published article.

**Abbreviations:** ANCOVA, Analyze of Covariance; AIDS, Acquired Immunodeficiency Syndrome; ART, Antiretroviral therapy; BMI, Body Mass Index; HBV, Human Biological Variability; HIV, Human immunodeficiency virus; MAC, Mid-arm Circumference; MVPA, Moderate-to-Vigorous Physical Activity; PA, Physical Activity; PAQ, Physical Activity Questioner; WHO, World Health Organization.

## Conclusion

The subjects participants in the study living with HIV and undergoing ART had impaired growth, low physical fitness and low levels of habitual physical activity in relation to the reference values of their peers without HIV, which compromised their physiological functioning and their quality of life.

## 1. Introduction

Since the discovery of the first cases of HIV in humans, in the 1980s, infection by HIV has been the focus of health care worldwide, and became one of the most serious contemporary issues of public health [1–3].

Before the advent of antiretroviral treatment (ART), children and adolescents with HIV had characteristics of protein-energy malnutrition and, as a consequence of this malnutrition, they had a high risk of mortality, reduced physical growth and muscle mass, delayed motor development, and reduced levels of physical activity and physical fitness, which substantially reduced their quality of life even when compared to their uninfected peers [4–11].

As a result of its effectiveness, the ART medicines resulted in viral suppression and partial restoration of the immune system of the infected individuals [12]. The introduction of ART has promoted a significant change in the natural course of the HIV infection, providing greater hope and improvement in the quality of life of infected persons [13].

However, ART has significant side-effects that counteract the positive health effects. Several side-effects were identified, caused by prolonged exposure to ART, promoting an early, slow and progressive metabolic process of complications that developed comorbidities, such as lipodystrophy, dyslipidemia, insulin resistance and reduced bone mineral density/content for age [14].

In Mozambique, ART was introduced to children in 2003 [15] and, since then, there has been an increase in the number of children and adolescents living with HIV as a chronic disease, as a consequence of the medication [16]. Although there are some complications related to the adherence of ART in the pediatric population worldwide, specific research in this population in Mozambique is unknown.

The effects related to HIV and ART are reported to affect the health-related fitness components such as body composition, physical fitness and physical activity [17–19]. The development of children and adolescents deserves attention and should be investigated as a prerequisite for creating strategies to maintain recommended health levels, which becomes even more important when it concerns children and adolescents living with chronic diseases, such as HIV and AIDS.

Consequently, the present was designed with the aim of determining the body composition, physical fitness and habitual physical activity of HIV-infected children and adolescents on ART. It is believed that the results of this study could make it possible to develop new health-care strategies for the treatment of children and adolescents living with HIV, which may help to simultaneously combat the two adverse phenomena of chronic HIV disease effects and the effects of ART drugs.

## 2. Methods

A cross-sectional study design was done with a convenient sample. Subjects were identified first by direct contact with the parents/guardian by the researchers, who were accompanied by

the healthcare practitioner on duty in the healthcare service centers for children and adolescents living with HIV in Maputo. The invitation for the subjects to participate in the study was made privately in a specific room at the healthcare centers for children and adolescents living with HIV, after their parents/guardian authorized and signed the informed consent form. The sample size was calculated based on a sampling error of 5% and a confidence level of 95% and was selected over a four-week period for each hospital, based on the need to try to capture the cycle of variability during the month (at the beginning, in the middle and at the end of the month). A total of 79 subjects (boys = 43; girls = 36), aged 8 to 14 years, participated in the study. All subjects were HIV positive, due to vertical transmission, and were regularly treated with ART since birth. The number of $CD4^+$ cells at the time of the tests was greater than or equal to 300 cells/μL, according to the medical information contained in the clinical file. The ART regimen in use and the viral load were not taken into account for this study. All subjects were clinically able to perform some amount of physical activity, and none were regular practitioners of physical exercise. The subjects underwent an evaluation consisting of anthropometry, physical fitness and daily physical activity.

### 2.1. Anthropometry

The anthropometric assessment included height, weight, mid-arm circumference (MAC) and skinfolds. The height was assessed through a stadiometer SECA model 2421814009 with the subjects barefoot. Body weight was assessed with a Secca scale, with subjects wearing light clothing and barefoot. Body weight and height were used to calculate the body mass index (BMI) using the formula BMI = weight/height$^2$. Tricipital and subscapular skinfolds were measured using a Hardpenden (model 10931) caliper. The sum of the two skinfolds was used to estimate excess body fat using the percentiles proposed by [20]. Relaxed mid-arm circumference (MAC) was obtained by anthropometric tape and evaluated according to the protocol described by [20].

### 2.2. Fitness tests

Motor performance included the flexibility of the trunk (sit-and- reach–[21]), abdominal strength/resistance (curl- up–[22]), handgrip strength (hand grip-[23]) that was measured using a manual dynamometer (GRIP-D TKK 5401) and lower limb power (standing long jump–[24]).

### 2.3. Physical activity

Physical activity was assessed with a pedometer (SC-StepRx®, Deep River, Canada), placed on the subject's waist region for 7 uninterrupted days, except for sleep and bathing.

### 2.4. Statistical analyses

Analyzes were performed using the SPSS program (version 22), with a significance level of p≤0.05. The normality of the data distribution was verified with the Kolmogorov Smirnov test. The proportions related to the nutritional status and the physical activity levels was determined by the Chi- square test, and the comparison of the subjects with and without the diagnosis of the disease used the ANCOVA test, with age as the covariate.

### 2.5. Reference data

For the classification of nutritional status and body composition [24], reference values for height/age and BMI/age were used. For the sum of the tricipital and subscapular skinfolds and arm circumference, the reference used was from [20].

For the fitness test, the reference values used were from [21], for trunk flexibility, it was from [22] and for abdominal strength/endurance, lower limb strength and handgrip strength, it was from [23].

Subjects were classified as either insufficiently active or active according to the average number of steps per day, using 12 000 steps/day as the cut-off point [25]. Classification of time in moderate-to-vigorous physical activity (MVPA) was based on the validated cutoff points of 110 steps/minute for moderate-intensity PA, and 130 steps/minute for vigorous-intensity PA [26]. Active subjects were those who performed 60 minutes or more PA per day.

To compare with non-HIV+ Mozambican peers of the same age and gender, studies were done in urban and rural areas in the project of Human Biological Variability (HBV) in Mozambique (Maputo, 2012 and Inhaca, 2019).

## 3. Results

### 3.1. Descriptive results

Descriptive results (n, mean±sd) from all anthropometric, fitness tests and physical activity are demonstrated in the S1–S4 Tables.

### 3.2. Anthropometry

Fig 1 shows the percentile position of each subject in relation to WHO (height/age and BMI/age) and in Frisancho (sum of skinfolds and MAC) standard reference values according to age and sex (Frisancho, 1990; WHO, 2007). According to height/age, most subjects were below the 50[th] percentile (93% for males and 91.6% for females), and 34.9% of boys and 27.8% of girls fell below the 5[th] percentile. Relative to BMI/age, the majority of subjects (boys = 72.1%; girls = 69.4%) fell below the 50[th] percentile, and there were only 5 cases of overweight (6.8%) and 6 cases of malnutrition (7. 6%). Most of the subjects in the sample had values below the 50[th] percentile for sum of skinfolds, which was more noticeable in females (75%) than in males (45%). Finally, the percentile distribution for MAC indicated that only 25.6% of boys and 25% of girls reached the 50[th] percentile, while 32.6% and 13.9% of boys and girls, respectively, presented with values below the 5[th] percentile.

Table 1 presents the anthropometric and body composition results of the comparison (ANCOVA) between HIV[+] subjects and non-HIV[+] Mozambican children, adjusted for age. Both HIV[+] boys and girls had lower values for height (p<0.000) in relation to non-HIV[+] boys and girls. Furthermore, BMI, sum of skinfolds and MAC in HIV[+] subjects were lower relative to an urban setting (p<0.000), but was non-significant for rural boys and girls.

Table 2 shows the comparison (%) related to nutritional status. HIV[+] subjects had significantly higher levels of stunting (boys = 34.9%; girls = 27.8%) than non-HIV[+] urban (boys = 9.8%; girls = 8.0%) and rural (boys = 31.5%; girls = 17.5%). Concerning tissue-wasting, the HIV[+] subjects had a higher proportion of girls (11.1%), but not boys (9.3%). The proportion of overweight and obesity was low (boys = 4.6%; girls = 2.7%) relative to urban, but not rural subjects.

### 3.3. Physical fitness test

Table 3 shows the proportion of HIV[+] subjects according fitness category. Most of the subjects were classified as unfit, whether boys or girls with the exception of the flexibility test (sit and reach).

The comparison of HIV[+] subjects with non-HIV+ Mozambican peers from urban and rural areas is illustrated in Table 4. There was no difference in flexibility (sit and reach)

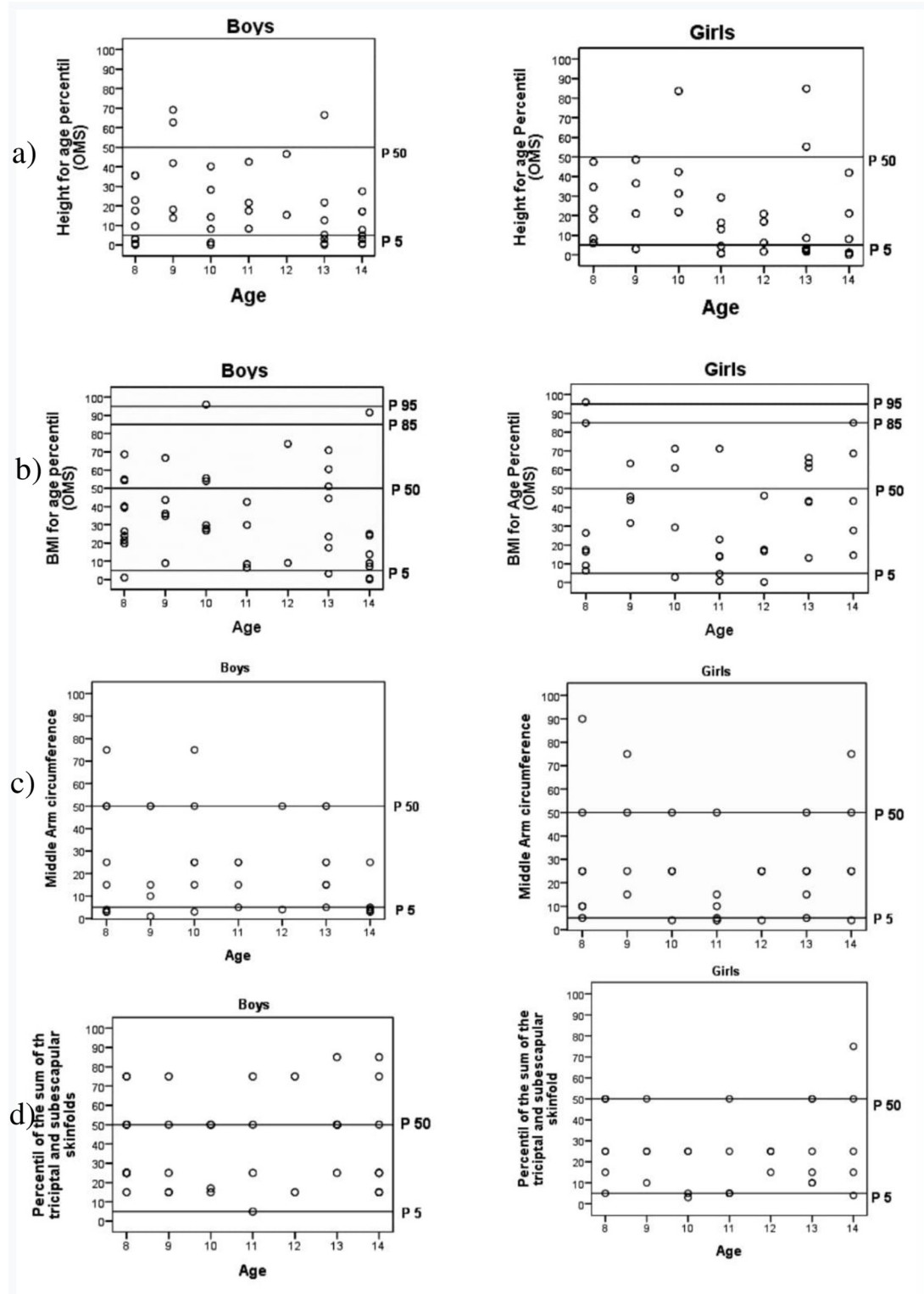

**Fig 1.** Individual position of each subject relative to reference standards in a) height/age and gender, b) BMI/age and gender, c) MAC/age and gender and d) sum of skinfolds/age and gender.

**Table 1. Comparision (mean±sd) of height, body mass index (BMI), sum of tricipital and subscapular skinfolds (∑skinfolds) and mid-arm circumference (MAC) between HIV+ and non-HIV+ Mozambicans boys and girls from urban and rural settings; p values were generated from an ANCOVA test with age as a covariate.**

| Variable | HIV+ | Urban | Rural | P |
|---|---|---|---|---|
| | | **Boys** | | |
| Height | 135.9 ± 1.2[a] | 142.7 ± 0.3[b] | 140 ± 0.5[c] | 0.000 |
| BMI | 16.6 ± 0.4[ab] | 17.5 ± 0.1[a] | 16.5 ± 0.2[b] | 0.000 |
| ∑Skinfolds | 13.5 ± 1.1[ab] | 14.1 ± 0.3[a] | 12.3 ± 0.5[b] | 0.005 |
| MAC | 19.7 ± 0.3[ab] | 20.5 ± 0.9[a] | 19.7 ± 0.1[b] | 0.000 |
| | | **Girls** | | |
| Height | 137.6 ± 1.3[a] | 143.8 ± 0.3[b] | 141.7 ± 0.5[c] | 0.000 |
| BMI | 16.9 ± 0.5[a] | 18.6 ± 0.1[b] | 17.3 ± 0.2[a] | 0.000 |
| ∑Skinfolds | 16.8 ± 1.9[ab] | 20.5 ± 0.4[a] | 16.3 ± 0.7[b] | 0.000 |
| MAC | 20.6 ± 0.4[ab] | 21.4 ± 0.1[a] | 20.8 ± 0.1[b] | 0.000 |

a) significantly different to b) and c);
b) significant different to a) and b);
c) significantly different to a) and c).

performance. Abdominal strength/endurance was significantly lower in HIV+ (boys = 11.3 ± 2.8 and girls = 9.8±3.1 repetitions/minute) subjects related to their urban and rural peers who were not living with HIV (p = 0.000). A similar result was observed for hand-grip strength (boys = 17.3 ± 0.7 and girls = 17.3 ± 0.7 kg). In standing long jump, the values were lower for urban subjects, but statistically identical for rural subjects.

## 3.4. Physical activity

The average number of steps in HIV+ children was 11 962±960 steps/day in boys and 9 805 ±818 steps/day in girls (Table 5). The time spent in MVPA was 48.9±5.4 min/day for boys and 38.7±4.9 min./day for girls. When compared to their urban and rural non-HIV+ peers, the differences were significant for both boys and girls whether for steps per day or for MVPA. In all cases, the HIV+ subjects spent less time in total physical activity and in MVPA.

The proportion classified as insufficiently active according to the number of steps per day (<12 000 steps/day) in HIV+ subjects were 46.5% in boys and 77.8% in girls (Table 6). This proportion was significantly higher than their Mozambican peers from both urban and rural areas. When using MVPA (<60 min/day), this proportion changed to 67.4% for boys and 91.7% for girls who were HIV+. Again, the proportion of participants who were insufficiently

**Table 2. Proportion (%) of stunting (low height for age), wasting (low weight for height) and overweight and obesity between HIV+ and non-HIV+ urban and rural Mozambican boys and girls from the Chi-square test.**

| Variable | HIV+ | Urban | Rural | Chi-square | P |
|---|---|---|---|---|---|
| | | **Boys** | | | |
| *Low height for age* | 34.9 | 9.8 | 19.2 | 31.5 | 0.000 |
| *Low weight for height* | 9.3 | 8.9 | 9.6 | 23.9 | 0.001 |
| *Overweight + obesity* | 4.6 | 10.8 | 1.4 | | |
| | | **Girls** | | | |
| *Low height for age* | 27.8 | 8.0 | 12.1 | 17.5 | 0.000 |
| *Low weight for height* | 11.1 | 5.6 | 5.9 | 49.072 | 0.000 |
| *Overweight + obesity* | 2.7 | 18.1 | 3.7 | | |

**Table 3. Proportion (%) of subjects by fitness category (fit and unfit).**

| Fitness Test | Fit | Unfit |
|---|---|---|
| | Boys | |
| Sit and reach | 83.7 | 16.3 |
| Curl-up | 27.9 | 72.1 |
| Hand grip | 18.6 | 81.4 |
| Standing long jump | 25.6 | 74.4 |
| | Girls | |
| Sit and reach | 91.7 | 8.3 |
| Curl-up | 19.4 | 80.6 |
| Hand grip | 30.6 | 69.4 |
| Standing long jump | 41.7 | 58.3 |

active was higher in the HIV[+] participants compared to the HIV negative urban and rural participants (Table 7).

# 4. Discussion

The study aimed to determine the body composition, physical fitness and physical activity of children and adolescents living with HIV on ART. The results in the present study suggest that children and adolescents with HIV undergoing ART, have impaired somatic growth, as well as decrements in some components of physical fitness. In essence, the data suggests that HIV[+] children and adolescents have lower levels of physical activity than their non-HIV positive peers.

## 4.1. Somatic variables

The somatic values in the HIV[+] participants fell below the 50[th] percentile of the WHO standards, which is associated with a reasonable prevalence of stunting and tissue-wasting (30.1% and 10.1%, respectively). In general, the comparison of somatic values with published

**Table 4. Comparison (mean±sd) of fitness tests for HIV[+] subjects in urban and rural Mozambican boys and girls; p values were generated from the ANCOVA test with age as a covariate.**

| Variables | HIV[+] | Urban | Rural | F |
|---|---|---|---|---|
| | Boys | | | |
| Sit and reach (cm) | 28.9 ± 0.9 | 31.2 ± 0.3 | 31.2 ± 0.4 | 0.070 |
| Curl-up (reps/min) | 11.3 ± 2.8[a] | 28.5 ± 0.8[b] | 25.0 ± 1.1[c] | 0.000 |
| Hand grip (kg) | 17.3 ± 0.7[a] | 20.9 ± 0.2[b] | 19.9 ± 0.3[c] | 0.000 |
| Standing long jump (cm) | 133.4 ± 3.1[a] | 146.8 ± 0.8[b] | 128.0 ±1.3[a] | 0.000 |
| | Girls | | | |
| Sit and reach (cm) | 33.2 ± 1.0 | 34.1 ± 0.2 | 34.8 ± 0.2 | 0.200 |
| Curl-up (reps/min) | 9.8± 3.1[a] | 26.6 ± 0.7[b] | 21.1 ± 1.2[c] | 0.000 |
| Hand grip (kg) | 17.3 ± 0.7[a] | 20.3 ± 0.1[b] | 19.5 ± 0.5[b] | 0.000 |
| Standing long jump (cm) | 126.1± 3.5[a] | 135.3 ± 0.8[b] | 120.1 ± 1.3[a] | 0.000 |

a) significantly different to b) and c);

b) significantly different to a) and b);

c) significantly different to a) and c).

**Table 5. Comparison (mean±sd) of average number of steps per day and time spent in MVPA between HIV[+] and HIV negative urban and rural Mozambican boys and girls; p values were generated from the ANCOVA test with age as a covariate.**

| Variable | HIV[+] | Urban | Rural | P |
|---|---|---|---|---|
| | | *Boys* | | |
| *Step count (steps/day)* | 11 962 ± 960[a] | 16 431 ± 359[b] | 17 442 ± 1151[b] | 0.000 |
| *MVPA (min/day)* | 48.9 ± 5.4[a] | 70.6 ± 2.0[b] | 68.8 ± 5.5[b] | 0.001 |
| | | *Girls* | | |
| *Step count (steps/day)* | 9 805 ± 818[a] | 12 492 ± 241[b] | 13 949 ± 818[b] | 0.001 |
| *MVPA (min/day)* | 38.7 ± 4.9 | 51.1 ± 1.5 | 51.9 ± 4.4 | 0.05 |

literature from other studies carried out on Mozambican children and adolescents of the same age and sex without a diagnosis of HIV reported that subjects with an HIV[+] status were at a disadvantage. This was observed both for urban and rural boys and girls, which leads to the suggestion that, somehow, being HIV[+] and/or on ART negatively affected the normal growth of HIV[+] children and adolescents.

This finding seems to be common to many studies, which states that children and adolescents living with HIV tend to be smaller, have lower body weight and enter puberty later compared to their apparently healthy peers [27–29].

Dos Reis et al. [30] in a study with Brazilian children with HIV, observed a high prevalence (20.9%) of stunting, a low prevalence of central adiposity and a depletion of adipose tissue (<5[th] percentile) about 33.9%. Similarly, Cardoso [31] identified the presence of low weight and low height for age, but in smaller proportions (6% and 8%, respectively) compared to those found in the present study. Furthermore Tanaka et al. [32], when evaluating 91 Brazilian adolescents with HIV, aged between 10 and 19 years, observed the high prevalence of low height for age and malnutrition in 15.4% and 9.9% respectively, while overweight / obesity was present in 12.1%.

There are multiple studies in children and adolescents with HIV that have reported negative Z scores for height and low height for age [33–38].

Ramalho and Da Silva [39] also found that a group of 50 Brazilian adolescents, of both sexes with HIV, had lower height compared to the control group that presented without the diagnosis of HIV infection. Similarly, Malete et al. [40] found in adolescents of Tswana descent, that the subjects without a diagnosis of HIV were taller in comparison with those living with HIV.

However, the determinants of these disease outcomes, also known as stunting and low weight for height, are complex and only partially understood, with a multifactorial source of origin [41]. Examples of these multifactorial factors are socioeconomic status, mitochondrial toxicity, psychosocial factors, deficiency in micronutrient intake and absorption, abnormal nitrogen balance and, amongst others, impaired growth hormone secretion [42].

Longitudinal follow-up studies [43, 44] have shown that after the initiation of ART, and consequent viral suppression, there was an improvement in the weight and height of the

**Table 6. Comparison (%) of subjects according to activity categories classified by the average number of steps per day; p values resulted from the Chi-square test.**

| Variable | HIV[+] | Urban | Rural | Chi–square | P |
|---|---|---|---|---|---|
| | | *Boys* | | | |
| *Insufficiently active* | *46.5* | *24.0* | *20.9* | *10.6* | *0.005* |
| *Active* | *53.5* | *76.0* | *79.1* | | |
| | | *Girls* | | | |
| *Insufficiently active* | *77.8* | *46.5* | *42.6* | *13.7* | *0.001* |
| *Active* | *22.2* | *53.5* | *57.4* | | |

**Table 7. Comparison of subjects according to activity categories classified by the average amount of time spent in MVPA; p values resulted from the Chi-square test.**

| Group | HIV+ | Urban | Rural | Chi–square | P |
|---|---|---|---|---|---|
| *Boys* | | | | | |
| *Insufficiently active* | 67.4% | 40.7% | 39.5% | 11.3 | 0.003 |
| *Active* | 32.6% | 59.3% | 60.5% | | |
| *Girls* | | | | | |
| *Insufficiently active* | 91.7% | 69% | 70.2% | 8.2 | 0.02 |
| *Active* | 8.3% | 31% | 29.8% | | |

infected individuals during the first and second years, respectively. However, the transversal character of the present study, as well as the limits of the selected variables and the sample size, do not allow one to determine to what extent the effect on growth found in these participants is due to the specific condition of being HIV+. Thus, a longitudinal follow-up study will be necessary in order to ascertain how growth occurs, its deficiencies and what other factors may interfere with this process [45]. Such research would be quite interesting and relevant to the present research, because the subjects would then present a time of exposure to ART considered sufficiently long and likely to cause various effects, some beneficial and others harmful.

## 4.2. Physical fitness

The physical fitness in the subjects with HIV was low, compared to the reference group [20–22]. With the exception of trunk flexibility, most boys and girls living with HIV were classified as unfit for abdominal strength (72.1% vs 80.6%), handgrip strength (81.4% vs 69, 4%) and lower limb power (74.4% vs 58.3%). Thus, both rural and urban individuals from Mozambique, who were HIV-free, had significantly higher fitness values, reflecting that HIV+ participants on ART had lower fitness levels. This was consistent across a number of fitness variables, with a few exceptions.

The complications of HIV infection related to deficits in physical fitness were reported in several studies [7, 46]. These deficits can, in turn, lead to a low quality of life in children and adolescents living with HIV, and an increased risk of reduced musculoskeletal function [47–49].

Most of the studies in the literature on children and adolescents living with HIV reveal similar results for physical fitness. Dos Santos [45] evaluated 63 Brazilian children and adolescents living with HIV and found that both abdominal muscle strength and flexibility reported values below the cutoff point in both sexes. Barros et al. [50] studied the muscular strength of 33 Brazilian children of both sexes, aged between 7 and 12 years, all with HIV on ART, and observed that the scores in the neuromotor tests were below the normal range, as well as in relation to the group of children who did not have the virus, even when compared by sex, which agrees with the results of the present study.

Also, in a study that evaluated the relationship between HIV infection and physical fitness, Somarriba [33] reported lower values for muscle strength and flexibility in North American children and youth of both sexes, aged between 7 and 20 years, when compared with the control group without infection. They emphasized that factors associated with the decrease in peak $\dot{V}O_2$, the increase in the percentage of total body fat and longer time on ART, in HIV infected subjects, can induce muscle and metabolic abnormalities that contribute to the decrease in motor performance.

Ramalho [41], in a cross-sectional study which evaluated children and adolescents with HIV on ART, aged 7 to 20 years, found that they had lower levels of physical fitness (horizontal

jump, flexion and elbow extension, sit-and-reach, and abdominal flexion) than their peers without infection.

Macdonald [37] investigated the muscle power of the lower limbs of Canadian subjects, aged 8 to 25 years, who acquired HIV in the perinatal period. They observed lower levels of muscle strength compared to individuals not infected with HIV. The authors suggested that HIV acquired in the perinatal period could be associated with deficits in muscle power, body mass, and muscle function that influenced the activities of daily living and the overall quality of life. However, it will be important to better understand the clinical significance of these differences.

This phenomenon may be due to the fact that, on the one hand, chronic conditions affecting health in childhood and adolescence usually limit participation in physical activity and sport, as a consequence of real or perceived limitations imposed by the disease, On the other hand, TARV induces muscle and metabolic abnormalities that contribute to the reduction in exercise performance [33, 65].

Different from these results, there are studies on children and adolescents living with HIV in which the results contradict this trend of a negative impact of HIV infection on physical fitness. Ramos et al. [51], for example, when studying Puerto Rican preadolescents with an average age of 11 years found that arm strength did not show significant differences between the HIV infected and non-infected participants,. Cardoso [31] also investigated the handgrip strength of 50 Brazilian individuals living with HIV, aged 8 to 15 years, and found that the average was in the middle region to the values found in studies with subjects considered healthy. However, the authors justify that these values may have been influenced by a series of factors, such as the methodology, the types of instruments used, and the small sample size in practically all of these studies.

The impact of body size and body composition on physical fitness is widely demonstrated [52–54]. Thus, it could be hypothesized that the inferior performance of children and adolescents with HIV infection was due to the low somatic measurements. However, in the present study, when introducing the variables of height and body mass index as covariates in the model of data analysis, the statistical differences remained significant.

The tendency towards reduced levels of motor performance in subjects with HIV can also be explained by limitations in the participation of activities of daily living, impaired strength and muscle tone, and reduced skeletal muscle mass and function that are associated with energy deficiencies caused by the infection itself and related to chronic inflammatory diseases [55–57]. Thus, intervention studies that can induce an increase in physical activity and physical exercise may contribute to clarifying this issue.

### 4.3. Physical activity

The subjects' physical activity was evaluated using a pedometer, from which the daily mean number of steps were counted, and the amount of time engaged in MVPA were considered. Contrary to the trend observed in pediatric Mozambican populations without HIV infection, where the observed levels of physical activity were higher in relation to the recommendations, [58], the observed results of this study showed that the majority of subjects (60.8%) did not reach the average number of steps recommended in the literature or (79.7%) the recommendations for daily time in MVPA [25]. Several authors have postulated that children and adolescents with chronic conditions generally have restrictions on participation in physical activity and sport, due to real and perceived limitations which, in the case of HIV infection, are characterized by a combination of factors, such as social stigma and dissatisfaction with body image, which contribute to low self-esteem and social seclusion, leading to a sedentary lifestyle [59–61].

The results found in the present study are in agreement with those of Cardoso [34] who analyzed the level of physical activity of 46 Brazilian children and adolescents of both sexes living with HIV using a pedometer and found that they had levels of physical activity below the recommended values, and below those of healthy children and adolescents.

Similar results were found by Ramalho and Silva [39], when comparing the levels of physical activity of 50 Brazilian HIV infected patients with 64 patients in the control group without a diagnosis of infection, aged 7 to 20 years, in which the majority (75%) of the HIV patients presented with a higher prevalence of physical inactivity, and less weekly time in physical activity, when compared with the control group. Consequently, the majority did not reach the minimum values of physical activity according to the WHO recommendations.

Martins [62], using the physical activity questionnaire (PAQ-C) evaluated 57 Brazilian adolescents of both sexes living with HIV, aged 10 to 15 years, and found that a large number (96.5%) of subjects were insufficiently active in relation to the recommendations for daily physical activity.

In a cross-sectional study by Martins et al. [38] with a sample of 57 Brazilian adolescents of both sexes with HIV, aged between 10 and 15 years, and a control group of the same size and age, evaluated their physical activity using the physical activity questionnaire (PAQ-C). It was observed that the adolescents with HIV had lower scores for physical activity when compared to their peers considered healthy.

Tanaka et al. [32] in a study that evaluated 91 Brazilian adolescents, aged between 10 and 19 years with HIV using the physical activity questionnaire, found that in 71.4% of the subjects, the median time spent in physical activity was below the cutoff point, and was even lower in females. This finding agrees with the results of the present study.

A review study [63] concluded that there was a high prevalence of adolescents living with HIV/AIDS who did not comply with the minimum recommendations for physical activity, suggesting that the promotion of programs or projects with trained professionals, for example, in schools, involving their parents, becomes fundamental for increasing their levels of physical activity and, possibly, decreasing chronic noncommunicable diseases at other stages of life.

The low levels of fitness and physical activity may be due to the fact that there is still a lot of taboo and stigmatization in this population and the consequent overprotection of the parents, preventing them from socializing with their peers and practicing activities and games specific for their age. Furthermore, the factors that—influence the behavioral habits and levels of physical activity of individuals are multiple and complex. Therefore, it is necessary to consider other factors that determine the usual practice of physical activity, such as environmental, sociodemographic, biological, psychological, cognitive, and sociocultural.

However, in contrast, other studies found no differences in the levels of physical activity between the subjects living with HIV and the control groups without the diagnosis of the disease or with the reference values. Such is the case for Csordas and Lazarotto [64] who investigated the clinical profile of Brazilian adolescents, aged between 13 and 19 years, with HIV-AIDS and low adherence to antiretroviral therapy. The authors found that, regardless of gender, there was a high proportion (55%) of adolescents classified as highly active. Also, Wong et al. [36] evaluated South African children, aged 5 to 10 years, and observed high levels of physical activity among schoolchildren with HIV who had started treatment early and were controlled by ART. These children had similar results when compared to their peers without a diagnosis of the disease, suggesting that the early diagnosis and treatment of HIV infection can help to minimize the effects of the disease and keep individuals at the levels of health and physical activity close to or even equal to those considered desirable. However, in the same study, the authors found that among girls, the time spent in vigorous physical activity was significantly less in those diagnosed as HIV+ when compared to their non-HIV peers, even when

adjusting for age. De Lima et al. [65] using accelerometry, investigated the physical activity of 130 Brazilian children and adolescents, aged between 8 and 15 years, 65 of whom had HIV and a similar number without the infection. The results showed that, although the bouts of physical activity were low in participants with HIV, the total values for physical activity were similar in both groups.

The controversy in the results of several studies is concerning. These differences may be due to the fact that different methods were used for the different studies to estimate and classify the levels of physical activity, combined with the fact that the samples were reduced, as is the case in the present study. Also, different study contexts can be taken into account.

The factors that contribute to the habits and levels of physical activity of different individuals are multiple and complex. Therefore, it is necessary to consider other factors that determine the usual practice of activity, such as environmental, sociodemographic, biological, psychological, cognitive, and sociocultural, among others [66]. For example, Mangona et al. [67] demonstrated that high levels of physical activity in adult women living with HIV and undergoing ART was associated with cultural and survival factors, and not only with the pathological and physical determinants of the disease.

## 5. Limitations

The fear of caregivers in relation to the concerns for children and adolescents living with HIV, led to the sample being convenient and reduced in size. The absence of information on the level of viral load, and the scheme of antiretroviral treatment in most patients' chips, constituted a limitation that led to the non -use of these data.

Regarding the small sample size, it became evident that more studies with larger samples and more variables should be carried out to more accurately assess the effect of HIV, as well as the treatment of antiretroviral therapy in children and adolescents. On the other hand, a longitudinal follow-up study can provide more tangible information about the effect of somatic growth and the evolution of fitness and physical activity in these individuals.

## 6. Conclusion

Children and adolescents living with HIV on ART are affected negatively in their somatic growth, physical fitness and habitual physical activity by a complex trait of factors that still remain to be clarified.

## Supporting information

**S1 Table. Descriptive data (mean±sd) of height, body mass index (BMI), sum of tricipital and subscapular skinfolds (Σskinfolds) and middle arm circumference (MAC).**
(DOCX)

**S2 Table. Descriptive data (mean±sd) of physical fitness tests of subjects with HIV.**
(DOCX)

**S3 Table. Descriptive data (mean±sd) of steps/day and average time spent in MVPA of subjects with HIV.**
(DOCX)

**S4 Table. Statistical procedures information.**
(DOCX)

## Author Contributions

**Conceptualization:** Nivaldo Chirindza.

**Data curation:** Lucília Mangona, Gomes Nhaca, Timóteo Daca.

**Formal analysis:** Nivaldo Chirindza, Lloyd Leach, António Prista.

**Investigation:** Nivaldo Chirindza, Lucília Mangona, Gomes Nhaca, Timóteo Daca.

**Methodology:** Nivaldo Chirindza.

**Project administration:** Nivaldo Chirindza.

**Supervision:** António Prista.

**Writing – original draft:** Nivaldo Chirindza.

**Writing – review & editing:** Nivaldo Chirindza, António Prista.

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
