## [Decision Letter · Decision Letter 0]

13 May 2022

PONE-D-21-23594BODY COMPOSITION, PHYSICAL FITNESS AND PHYSICAL ACTIVITY IN MOZAMBICAN CHILDREN AND ADOLESCENTS WITH HIVPLOS ONE

Dear Dr. Chirindza,

Thank you for submitting your manuscript to PLOS ONE. After careful consideration, we feel that it has merit but does not fully meet PLOS ONE’s publication criteria as it currently stands. Therefore, we invite you to submit a revised version of the manuscript that addresses the points raised during the review process.

We look forward to receiving your revised manuscript.

Kind regards,

Giordano Madeddu

Academic Editor

PLOS ONE

Journal Requirements:

3. In your Methods section, please provide a justification for the sample size used in your study, including any relevant power calculations (if applicable).

4. Thank you for stating the following financial disclosure: "We have not any grant, sponsorship or support from our institution to pay the publication we are requesting fee waive for the publication"

5. Thank you for stating the following in your Competing Interests section: "All data generated or analyzed during this study are included in this published article."

8. Please upload a new copy of Figure 1 as the detail is not clear. Please follow the link for more information: https://blogs.plos.org/plos/2019/06/looking-good-tips-for-creating-your-plos-figures-graphics/" https://blogs.plos.org/plos/2019/06/looking-good-tips-for-creating-your-plos-figures-graphics/

Reviewers' comments:

Reviewer's Responses to Questions

**Comments to the Author**

1. Is the manuscript technically sound, and do the data support the conclusions?

Reviewer #1: Yes

Reviewer #2: Partly

Reviewer #3: Yes

2. Has the statistical analysis been performed appropriately and rigorously? 

Reviewer #1: Yes

Reviewer #2: No

Reviewer #3: Yes

3. Have the authors made all data underlying the findings in their manuscript fully available?

Reviewer #1: Yes

Reviewer #2: Yes

Reviewer #3: Yes

4. Is the manuscript presented in an intelligible fashion and written in standard English?

Reviewer #1: Yes

Reviewer #2: Yes

Reviewer #3: Yes

5. Review Comments to the Author

Reviewer #1: General comment: The present study has a relevant theme and based on the findings of this study, health promotion actions can be better designed for children and adolescents with HIV.

Comment: In general the introduction is well written. Only at the end, the authors could add how the investigation of this information could help health professionals.

Comment: In the methodology, I suggest to the authors the inclusion of the sample size calculation, even if it is performed a posteriori.

Comment: In the methodology explain better how the children and adolescents who took part in the study were recruited.

Comment: In the statistical analysis, shouldn't sex be considered as an adjustment in ANCOVA?

Comment: When comparing variables between boys and girls, enter the P values showing whether there are significant differences.

“Most of the studies in the literature reveal identical results. Dos Santos [44] evaluated 63 Brazilian children and adolescents and found that both abdominal muscle strength and flexibility presented values below the cutoff point in both sexes. Barros et al. [49] studied the muscular strength of 33 Brazilian children of both sexes, aged between 7 and 12 years, all with HIV on ART, and observed that the scores in the neuromotor tests were below the normal range, as well as in relation to the group of children who did not have the virus, even when compared by sex, wich agree with the results of the present study.”

Comment: What would be the reasons? The authors seek to discuss the possible mechanisms.

“In a cross-sectional study Martins et al. [37] done with a sample of 57 Brazilian adolescents of both sexes between 10 and 15 years, with HIV and a control group of the same size and age and sexual level, using the physical activity questionnaire (PAQ-C). It was observed that adolescents with HIV had lower scores of physical activity when compared to their peers considered healthy.”

Comment: Why does this happen? What are the possible differences?

Comment: In the discussion, the authors make a series of important comparisons, but there is still a need to better discuss the reason for the differences or similarities of these comparisons.

Comment: A limitation that must be considered is the fact that the sample was by convenience (which does not diminish the importance of the study).

Comment: After the limitation paragraph, insert the study's strengths paragraph.

Comment: What are the practical applications of this study. How can this information help health professionals who care for children and adolescents with HIV?

Reviewer #2: The authors presented an interesting study about body composition and physical activity in Mozambican children and adolescents.

Although the article is well written and shows interesting conclusions, it fails to provide innovations. Per the authors admission, "This finding seems to be common to many studies, which state that children and adolescents

living with HIV tend to be smaller, have lower body weight and enter puberty later compared

to their apparently healthy peers".

Moreover, we found some serious problems the authors should address:

1) METHODS

a) "The ART regimen in use and the viral load were not taken into account for this study"

We find that this choice is problematic and limits the validity of the results of the study for several reasons. First of all, it is known that different drugs have different effects on metabolism, and especially lipid metabolism. Moreover, a persistent high level of plasma viral load leads to poor growth rates, due to persisting inflammation. In addition, low level viremia has been also related to persisting high levels of inflammation.

Therefore, we find that not considering actual ART regimen, past ART regimens (if any), time spent on ART regiment (present and past), and pVL makes the results of this study unreliable and non-reproducible.

2) RESULTS

a) "Descriptive results (n, mean +/- sd)"

Were the quantitative variables normally distributed? If not, "median (IQR)" should be used to summarize the variables.

Reviewer #3: Comments to the authors

PONE-D-21-23594

Title of the paper: Body composition, physical fitness and physical activity in children and adolescents with HIV.

The author(s) presented an interesting cross-sectional data on “Body composition, physical fitness and physical activity in children and adolescents with HIV from Mozambique”. There are some minor points which have to be considered in the manuscript.

Generally, the authors stated that ‘the adolescents and children with’; and this sounds not good and need to rephrased to include the word ‘living’ with, and this need to be addressed throughout the manuscript.

Title: The title is missing the inclusion of the word ‘living’ between the words ‘…adolescents .. and … with ..’. so that it reads as ‘Body composition, physical fitness and physical activity in children and adolescents living with HIV’.

Abstract: In the background and methods section first sentence, the authors are advised to reconsider the inclusion of the word ‘living’ as per comments above. In the second sentence the authors wrote a prefix for physical in capital letter and it is inconsistent with others, a clarification will be helpful.

The first conclusion sentence need to be rephrased, and it can be rephrased as ‘The subjects participants in the study living with HIV and undergoing ART had impaired growth………….’

Introduction

Can the authors clarify why the prefix for ‘development’ the sentence reading as ‘The Development of children…..’ written in capital letter.

In strengthening the arguments in the second paragraph of the introduction it would have been good if the authors could have considered a systematic review and meta-analysis by Rafaela Catherine da Silva Cunha de Medeiros and colleagues as published in 2021.

Methods

Can the authors clarify the how the contacts were done in the statement ‘Subjects were selected by contacts made with those who attended specific health care centers for children and adolescents with HIV in Maputo’ consideration the issues of stigmatisation amongst and ethical issues. It will helpful for the paper if the author could give a full description of the care centers settings in Mozambique. Also, the authors should provide the ethical approval number of the study given the vulnerability of the subjects participants in the study.

Under statistical analysis, sentence reading as ‘the proportion……’; an inconsistency has been observed in terms of the writing of sentences. As such corrections is required.

Spelling mistake for the word ‘same’ in the sentence reading as ‘To compare with non-HIV ± Mozambican peers of the some age and gender…..’. The authors are advised to make corrections.

Results

In table 1 and its text the authors for the first time are presented the distributions of the participants according to urban and rural settings of which such description is not explicit in the description of the participants in the methods. The authors should provide the description of their subjects participants in the methods section.

As indicated in Table 4 that it is about comparison of the HIV± and HIV-; can the authors clarify the distribution of this groups from their 79 included participants because it is unclear. Such clarification will helpful in understanding the subsequent analysis.

Discussion

In line with the comments regarding comparison between HIV± and HIV-, the major findings outlined in the first paragraph remain sketchy until a clarification is made about the distribution of the participants in accordance with this distribution.

Second paragraph, the name of the country Mozambique is written in small letter, as such corrections is needed.

Page 18, sentence reading as ‘This was consistent with few exception’. What are those exceptions, because this sentence seems lacking supporting statements. Clarification is required.

Under the subheading ‘Physical fitness’; 2nd paragraph sentence reading as ‘…..compared by sex, wich agree with the results of the present study.’ There is a spelling mistake for the word ‘which’; and corrections is needed.

The described limitations are without a proper link with the results and how they affected the current findings. As such, a more concise clarification regarding the limitations is required.

References

Generally, there are inconsistence references whereby some date are written in brackets (i.e. Ref numbers; 7;28;48; 55) and others not or just after the authors; and or at the end of the journal name.

6. PLOS authors have the option to publish the peer review history of their article (what does this mean?). If published, this will include your full peer review and any attached files.

Reviewer #1: No

Reviewer #2: No

Reviewer #3: No

---

## [Author Response · Author response to Decision Letter 0]

1 Aug 2022

It is an honor to be able to submit the manuscript to your scientific journal. Peer reviews make us learn a little more about manuscript preparation. I sincerely hope that this manuscript will be accepted for publication.

Review Comments to the Author

Reviewer #1: General comment: The present study has a relevant theme and based on the findings of this study, health promotion actions can be better designed for children and adolescents with HIV.

Comment: In general, the introduction is well written. However, at the end, the authors have added how the study could help healthcare professionals.

Answer: We added as reviewer's recommendation; “It is believed that the results of this study could make it possible to develop new healthcare strategies for the treatment of children and adolescents living with HIV, which may help to simultaneously combat the two adverse phenomena of chronic HIV disease effects and the effects of ART drugs.”

Comment: In the methodology, I suggest to the authors the inclusion of the sample size calculation, even if it is performed a posteriori.

Answer: We included as the recommendations: “The sample size was calculated based on a sampling error of 5% and a confidence level of 95% and was selected over a four-week period for each hospital, based on the need to try to capture the cycle of variability during the month (at the beginning, in the middle and at the end of the month).” 

Comment: In the methodology explain better how the children and adolescents who took part in the study were recruited.

Answer: We described in the text according to the reviewer's recommendations: “The invitation for the subjects to participate in the study was made privately in a specific room at the healthcare centres for children and adolescents living with HIV, after their parents/guardian authorized and signed the informed consent form.”

Comment: In the statistical analysis, shouldn't sex be considered as an adjustment in ANCOVA?

Answer: We made the statistical calculations separately from sex. So that sex was not used for adjustment.

Comment: When comparing variables between boys and girls, enter the P values showing whether there are significant differences.

Answer: In this study, we didn’t compare subjects by sex, but by urban and rural settings.

“Most of the studies in the literature revealed identical results. Dos Santos [44] evaluated 63 Brazilian children and adolescents and found that both abdominal muscle strength and flexibility presented values below the cutoff point in both sexes. Barros et al. [49] studied the muscular strength of 33 Brazilian children of both sexes, aged between 7 and 12 years, all with HIV on ART, and observed that the scores in the neuromotor tests were below the normal range, as well as in relation to the group of children who did not have the virus, even when compared by sex, which agreed with the results of the present study.”

Comment: What would be the reasons? The authors seek to discuss the possible mechanisms. 

Answer: We did discuss it: “This phenomenon may be due to the fact that, on the one hand, chronic conditions, such as HIV and AIDS, affecting health in childhood and adolescence usually limit participation in physical activity and sport, as a consequence of real or perceived limitations imposed by the disease. On other hand, TARV treatment induces muscle and metabolic abnormalities that contribute to the reduction in exercise performance.”

“In a cross-sectional study by Martins et al. [37] with a sample of 57 Brazilian adolescents of both sexes, between 10 and 15 years, with HIV, and a control group of the same size and age and sexual level, using the physical activity questionnaire (PAQ-C). it was observed that adolescents living with HIV had lower scores of for physical activity, when compared to their peers who were considered healthy.”

Comment: Why does this happen? What are the possible differences?

Answer: The low levels of fitness and physical activity may be due to the fact that there is still a lot of taboo and stigmatization in this population and the consequent overprotection of the parents, preventing them from socializing with their peers and practicing activities and games specific for their age. Furthermore, the factors that influence the behavioral habits and levels of physical activity of individuals are multiple and complex. Therefore, it is necessary to consider other factors that determine the usual practice of physical activity, such as environmental, sociodemographic, biological, psychological, cognitive, and sociocultural.

Comment: In the discussion, the authors make a series of important comparisons, but there is still a need to better discuss the reason for the differences or similarities of these comparisons.

Answer: The authors paid attention to the reviewer's comment and revised as recommended.

Comment: A limitation that must be considered is the fact that the sample was by convenience (which does not diminish the importance of the study).

Answer: The change was considered according to the reviewer's suggestion.

Comment: After the limitation paragraph, insert the study's strengths paragraph.

Answer: this was done as recommended by the reviewer.

Comment: What are the practical applications of this study. How can this information help health professionals who care for children and adolescents with HIV?

Answer: This is described in the last paragraph of the introduction.

Reviewer #2: The authors presented an interesting study about body composition and physical activity in Mozambican children and adolescents.

Although the article is well written and shows interesting conclusions, it fails to provide innovations. Per the authors admission, "This finding seems to be common to many studies, which state that children and adolescents living with HIV tend to be smaller, have lower body weight and enter puberty later compared to their apparently healthy peers".

Moreover, we found some serious problems the authors should address:

1) METHODS

a) "The ART regimen in use and the viral load were not taken into account for this study"

We find that this choice is problematic and limits the validity of the results of the study for several reasons. First of all, it is known that different drugs have different effects on metabolism, and especially lipid metabolism. Moreover, a persistent high level of plasma viral load (pVL) leads to poor growth rates, due to persisting inflammation. In addition, low level viremia has been also related to persisting high levels of inflammation.

Therefore, we find that not considering actual ART regimen, past ART regimens (if any), time spent on ART regiment (present and past), and pVL makes the results of this study unreliable and non-reproducible.

Answer: As we mentioned, the absence of information about the level of the viral load, the amount of TCD4 and the scheme of antiretroviral treatment in most individual patients constituted a limitation that led to the non -use of these data.

2).RESULTS

a) "Descriptive results (n, mean +/- sd)"

Were the quantitative variables normally distributed? If not, "median (IQR)" should be used to summarize the variables.

Answer: Yes. The quantitative variables were normally distributed.

Reviewer #3: Comments to the authors

Title of the paper: Body composition, physical fitness and physical activity in children and adolescents with HIV.

The author(s) presented an interesting cross-sectional data on “Body composition, physical fitness and physical activity in children and adolescents with HIV from Mozambique”. There are some minor points which have to be considered in the manuscript.

Generally, the authors stated that ‘the adolescents and children with’; and this sounds not good and need to rephrased to include the word ‘living’ with, and this need to be addressed throughout the manuscript.

Title: The title is missing the inclusion of the word ‘living’ between the words ‘…adolescents .. and … with ..’. so that it reads as ‘Body composition, physical fitness and physical activity in children and adolescents living with HIV’.

Answer: The word "living" was added as recommended by the reviewer

Abstract: In the background and methods section first sentence, the authors are advised to reconsider the inclusion of the word ‘living’ as per comments above. In the second sentence the authors wrote a prefix for physical in capital letter and it is inconsistent with others, a clarification will be helpful.

The first conclusion sentence needs to be rephrased, and it can be rephrased as ‘The subjects participants in the study living with HIV and undergoing ART had impaired growth………….’

Answer: The sentence has been reformulated according to the reviewer's recommendations

Introduction

Can the authors clarify why the prefix for ‘development’ the sentence reading as ‘The Development of children…..’ written in capital letter.

In strengthening the arguments in the second paragraph of the introduction it would have been good if the authors could have considered a systematic review and meta-analysis by Rafaela Catherine da Silva Cunha de Medeiros and colleagues as published in 2021.

Answer: The initial "D" was corrected by "d" in that word, and the systematic review and meta-analysis by Rafaela Catherine da Silva Cunha de Medeiros and colleagues as published in 2021 had been considered.

Methods

Can the authors clarify the how the contacts were done in the statement ‘Subjects were selected by contacts made with those who attended specific health care centers for children and adolescents with HIV in Maputo’ consideration the issues of stigmatization amongst and ethical issues. It will helpful for the paper if the author could give a full description of the care centers settings in Mozambique. Also, the authors should provide the ethical approval number of the study given the vulnerability of the subjects/participants in the study.

Answer: The way in which the subjects were contacted is described in the methodology. “Subjects were selected first by direct contact with the guardian by the researchers, who was accompanied by the health care practitioner on duty. Direct contact s made with those (parents/guardians) who attended specific health in care service centers for children and adolescents living with HIV in Maputo. The invitation to participate in the study for the subjects was made in a specific room for the care of children and adolescents with HIV, only made after their parents authorized and signed the informed consent form.”

The ethical approval number of the study was included in the end of the paper after author’s contributions according to the standards for authors of PLOS ONE.

Under statistical analysis, sentence reading as ‘the proportion……’; an inconsistency has been observed in terms of the writing of sentences. As such corrections is required.

Answer: We corrected as recommended 

Spelling mistake for the word ‘same’ in the sentence reading as ‘To compare with non-HIV ± Mozambican peers of the some age and gender…..’. The authors are advised to make corrections.

The Spelling mistake for the word ‘same’ was corrected as recommended.

Results

In table 1 and its text the authors for the first time are presented the distributions of the participants according to urban and rural settings of which such description is not explicit in the description of the participants in the methods. The authors should provide the description of their subjects/participants in the methods section.

As indicated in Table 4 that it is about comparison of the HIV+ ± and HIV-; can the authors clarify the distribution of this groups from their 79 included participants because it is unclear. Such clarification will helpful in understanding the subsequent analysis.

Answer: As indicated in the reference data chapter, to compare with non-HIV+ Mozambican peers of the same age and gender (and indicated how many participants were used), the authors used data of studies that were done in urban and rural areas in the project of Human Biological Variability (HBV) in Mozambique, (Maputo, 2012 and Inhaca, 2019).

Discussion

In line with the comments regarding comparison between HIV± and HIV-, the major findings outlined in the first paragraph remain sketchy until a clarification is made about the distribution of the participants in accordance with this distribution.

Answer: This is clarified in the answer above.

Second paragraph, the name of the country Mozambique is written in small letter, as such corrections is needed.

Answer: In the paragraph in question, it is not written Mozambique, but “Mozambican”.

Page 18, sentence reading as ‘This was consistent with few exceptions’. What are those exceptions, because this sentence seems lacking supporting statements. Clarification is required.

Answer: This part has been removed in order to resolve the inconsistency.

Under the subheading ‘Physical fitness’; 2nd paragraph sentence reading as ‘…..compared by sex, which agree with the results of the present study.’ There is a spelling mistake for the word ‘which’; and corrections is needed.

Answer: We corrected the spelling mistake as recommended.

The described limitations are without a proper link with the results and how they affected the current findings. As such, a more concise clarification regarding the limitations is required.

Answer: We re-wrote the limitations according the reviewer recommendation 

References

Generally, there are inconsistence references whereby some date are written in brackets (i.e. Ref numbers; 7;28;48; 55) and others not or just after the authors; and or at the end of the journal name.

Answer: We fixed the references according to the guidelines.

---

## [Decision Letter · Decision Letter 1]

27 Sep 2022

BODY COMPOSITION, PHYSICAL FITNESS AND PHYSICAL ACTIVITY IN MOZAMBICAN CHILDREN AND ADOLESCENTS LIVING WITH HIV

PONE-D-21-23594R1

Dear Dr. Chirindza,

We’re pleased to inform you that your manuscript has been judged scientifically suitable for publication and will be formally accepted for publication once it meets all outstanding technical requirements.

Kind regards,

Diego Augusto Santos Silva, Ph.D.

Academic Editor

PLOS ONE

Additional Editor Comments (optional):

Reviewers' comments:

Reviewer's Responses to Questions

**Comments to the Author**

1. If the authors have adequately addressed your comments raised in a previous round of review and you feel that this manuscript is now acceptable for publication, you may indicate that here to bypass the “Comments to the Author” section, enter your conflict of interest statement in the “Confidential to Editor” section, and submit your "Accept" recommendation.

Reviewer #1: All comments have been addressed

Reviewer #3: All comments have been addressed

2. Is the manuscript technically sound, and do the data support the conclusions?

Reviewer #1: Yes

Reviewer #3: Yes

3. Has the statistical analysis been performed appropriately and rigorously? 

Reviewer #1: Yes

Reviewer #3: Yes

4. Have the authors made all data underlying the findings in their manuscript fully available?

Reviewer #1: Yes

Reviewer #3: Yes

5. Is the manuscript presented in an intelligible fashion and written in standard English?

Reviewer #1: Yes

Reviewer #3: Yes

6. Review Comments to the Author

Reviewer #1: The authors responded point by point to all my comments, as well as added the suggestions in the text. So I have no further comments to make.

Reviewer #3: I am satisfied with the revised manuscript as the authors addressed all the comments which was raised from the initial review.

7. PLOS authors have the option to publish the peer review history of their article (what does this mean?). If published, this will include your full peer review and any attached files.

Reviewer #1: **Yes: **Diego G.D. Christofaro- São Paulo State University (UNESP)

Reviewer #3: No

---

## [Editor Report · Acceptance letter]

10 Oct 2022

PONE-D-21-23594R1 

Body composition, physical fitness and physical activity in Mozambican children and adolescents living with HIV 

Dear Dr. Chirindza:

I'm pleased to inform you that your manuscript has been deemed suitable for publication in PLOS ONE. Congratulations! Your manuscript is now with our production department. 

Kind regards, 

on behalf of

Dr. Diego Augusto Santos Silva 

Academic Editor

PLOS ONE